# Can the two-parameter recursive digital filter baseflow separation method really be calibrated by the conductivity mass balance method?

Weifei Yang[1,2,3,4], Changlai Xiao[1,2,3,4], Zhihao Zhang[1,2,3,4], Xiujuan Liang[1,2,3,4]

[1] Key Laboratory of Groundwater Resources and Environment (Ministry of Education), Jilin University, Changchun 130021, PR China.

[2] Jilin Provincial Key Laboratory of Water Resources and Environment, Jilin University, Changchun 130021, PR China.

[3] National-Local Joint Engineering Laboratory of In-situ Conversion, Drilling and Exploitation Technology for Oil Shale, Jilin University, Changchun 130021, PR China.

[4] College of New Energy and Environment, Jilin University, No. 2519, Jiefang Road, Changchun 130021, PR China.

Correspondence to: Xiujuan Liang (xjliang@jlu.edu.cn); Changlai Xiao (xcl2822@126.com)

**Abstract:** The two-parameter recursive digital filter method (Eckhardt) and the conductivity mass balance method (CMB) are two widely-used baseflow separation methods favored by hydrologists. Some divergences in the application of these two methods have emerged in recent years. Some scholars believe that deviation of baseflow separation results of the two methods is due to uncertainty of the parameters of the Eckhardt method, and that the Eckhardt method should be corrected by reference to the CMB method. However, other scholars attribute the deviation to the fact that they contain different transient water components. This study aimed to resolve this disagreement by analyzing the effectiveness of the CMB method for correcting the Eckhardt method through application of the methods to 26 basins in the United States by comparison of the biases between the generated daily baseflow series. The results showed that the approach of calibrating the Eckhardt method against the CMB method provides a "false" calibration of total baseflow by offsetting the inherent biases in the baseflow sequences generated by the two methods. The baseflow sequence generated by the Eckhardt method usually includes slow interflow and bank storage return flow, whereas that of the CMB method usually includes high-conductivity water flushed from swamps and depressions by rainfall, but not low-conductivity interflow and bank storage return flow. This difference results in obvious peak misalignment and periodic deviation between the baseflow sequences obtained by the two methods, thereby preventing calibration. However, multi-component separation of streamflow can be achieved through comparison. Future research should

recognize the deviations between the separation results obtained by the different methods, identify the reasons for these differences, and explore the hydrological information contained therein.

## 1 Introduction

Streamflow usually contains components originating from different sources, such as surface runoff, interflow, groundwater runoff, bank storage return flow and water flushed out from wetlands or depressions by rainfall (Cartwright et al., 2014; Schwartz, 2007; McCallum et al., 2010; Lin et al., 2007). These components from different sources are usually characterized by different residence times and chemical and isotopic characteristics (Cartwright et al., 2018). Collectively, these components control the runoff process and water chemistry characteristics of a river, and consequently affect the ecosystem along the river (Howcroft et al., 2019; Saraiva Okello et al., 2018). For example, the existence of transient water sources (interflow and bank storage return flow) will results in different concentration vs. discharge relationships during the rising and falling limbs, thus forming a hysteresis loop (Cartwright et al., 2014; Cartwright and Miller, 2021). Quantitative estimation of the relative proportions and the temporal resolutions of these components is a prerequisite for accurately predicting hydrological processes and protecting the river ecosystem (Duncan, 2019).

Since it is almost impossible to directly measure the different components of streamflow, they are usually indirectly determined through separation of the runoff process (Hagedorn, 2020; Lin et al., 2007). Since the task of accurately separating the runoff process into individual components presents a difficult challenge, hydrologists generally separate streamflow into two components, namely surface runoff and baseflow (Chapman, 1999; Eckhardt, 2005; Schwartz, 2007; Tallaksen, 1995). Surface runoff is the rapid flow that occurs on the catchment surface during a rainfall event, whereas baseflow is long-term "slow" runoff regulated by regional groundwater and other delayed stores of water, such as interflow and bank storage return flow. (Cartwright et al., 2018; Cartwright and Miller, 2021; Nathan and McMahon, 1990). However, some recent studies have taken the approach of simply generalizing the baseflow to be equivalent to the regional groundwater discharge, thereby ignoring the influence of transient water sources (Lott and Stewart, 2016; Lyu et al., 2020; Stewart et al., 2007; Zhang et al., 2017). Using

the above generalization, hydrologists have developed a variety of two-component hydrograph separation methods, also called "baseflow separation methods", which have been reviewed in detail by Nathan and McMahon (1990) and Chapman (1999). These methods can be broadly placed into three categories: (1) graphical methods; (2) filtering methods, and; (3) mass balance

methods (Lott and Stewart, 2016; Rammal et al., 2018; Xie et al., 2020; Hagedorn, 2020; Yang et al., 2019c). Graphical methods, such as the hydrograph separation program (HYSEP) (Sloto and Crouse, 1996), United Kingdom Institute of Hydrology (UKIH) (Piggott et al., 2005) and streamflow partitioning method (PART) (Rutledge, 1998) are automated versions of the traditional manual separation method. The majority of baseflow series generated by these methods consist of broken lines and do not reflect the natural transition of baseflow (Duncan, 2019; Eckhardt, 2008).

Lyne and Hollick (1979) proposed the earliest single parameter filtering algorithm based on the principle of signal processing, which typically requires multiple forward and backward filtering. There is a great deal of randomness associated with the determination of the number of filtering passes, which often increases uncertainty in the separation results. Boughton (1993) proposed an improved two-parameter filtering algorithm following which Jakeman and Hornberger (1993) proposed a three-parameter filtering algorithm; however, no clear physical explanations for the parameters of either algorithm were provided

(Chapman, 1999). Eckhardt (2005) derived a new two-parameter recursive filtering algorithm, referred to as the Eckhardt method in the present study, based on the linear reservoir theoretical framework. The two parameters included in this algorithm are the recession coefficient ($\alpha$) reflecting the recession characteristics of baseflow and the $BFI_{max}$ reflecting the long-term baseflow proportion. Although the recession coefficient ($\alpha$) of the Eckhardt method can be easily determined by recession analysis, empirical analysis is required to determine the $BFI_{max}$ (Eckhardt, 2012). Collischonn and Fan (2013) proposed a

method to estimate $BFI_{max}$ based on the linear reservoir theory. The Eckhardt method has been widely applied due to its clear physical basis and easy operation (Guzmán et al., 2015; Hagedorn, 2020; Li et al., 2014; Xie et al., 2020; Zhang et al., 2017). However, some scholars have pointed out that there is a certain amount of uncertainty associated with the selection of the $BFI_{max}$ value in the Eckhardt method which requires correction by other methods (Lott and Stewart, 2016; Rammal et al., 2018; Saraiva Okello et al., 2018; Zhang et al., 2013), such as the mass balance method based on environmental tracers.

The mass balance method calculates the proportions of total streamflow of different streamflow components based on the distinct chemical compositions of these components (Blumstock et al., 2015; Genereux, 1998; Hagedorn, 2020). Since a single tracer can only separate two flow components, achieving the separation of multiple flow components requires the concurrent use of multiple different tracers. The earliest tracers used included inert ions such as chloride, stable isotopes and radioactive isotopes (Burns, 2002; Genereux, 1998; Stewart et al., 2007). The heavy resource costs of water sample collection and detection

limits the application of the mass balance method for long sequence baseflow separation. Alternately, hydrologists proposed a mass balance method using conductivity as a tracer (CMB), thereby reducing the cost of the mass balance method and facilitating the application of the method to separation of long sequence baseflow (Pinder and Jones, 1969; Yu and Schwartz, 1999; Stewart et al., 2007). The CMB method has since been widely applied by researchers (Cartwright et al., 2014; Hagedorn, 2020; Kronholm and Capel, 2015; Lott and Stewart, 2013; Lott and Stewart, 2016; Lyu et al., 2020; Miller et al., 2014; Saraiva

Okello et al., 2018; Yang et al., 2019a; Zhang et al., 2013).

The majority of studies have concluded that the tracer-based CMB method has a clear physical basis and is therefore one of the most objective baseflow separation methods. The CMB method is therefore often used as a reference within the analysis of the effects of other baseflow separation methods or within the correction of the parameters of other methods (Lott and Stewart, 2013; Lott and Stewart, 2016; Saraiva Okello et al., 2018; Stewart et al., 2007; Zhang et al., 2017). Stewart et al.

(2007) applied the CMB method for the correction of the window length of the HYSEP method, whereas Lott and Stewart (2016) used the CMB method within the correction of the $BFI_{max}$ parameter of the Eckhardt method so as to obtain BFI values or cumulative baseflow consistent with that of the CMB method. Zhang et al. (2013) used the CMB method to correct two parameters of the Eckhardt method. Saraiva Okello et al. (2018) used discrete conductivity values for the correction of the $BFI_{max}$ of the Eckhardt method. The main objective of these corrections was to obtain a consistent BFI or cumulative baseflow,

but they do not spend too much time analyzing the fitting degree or deviation of daily baseflow. In other words, these studies only proved that the CMB method can be used to correct the BFI separated by the Eckhardt method on a multi-year scale, but could not explain the effect of calibration on the intra-year (daily) scale. However, other studies have found that the baseflow

sequence generated by the CMB method may contain different flow components compared to that generated by the Eckhardt method (Cartwright et al., 2014; Rammal et al., 2018). The correction of the Eckhardt method based on the CMB method should only be performed under the condition of both methods containing the same flow components (Hagedorn, 2020). Cartwright et al. (2014) analyzed the contribution of different sources of water to streamflow by comparing the differences in the results of baseflow separation by the Eckhardt and CMB methods. Since these two types of applications appear to be quite different, their join application can cause great confusion to hydrologists, mainly related to whether they can be used for calibration or comparison, and weather calibration on the daily scale is truly possible. As mentioned above, streamflow consists of many flow components, which are generalized into two broad types for the convenience of separation. The ultimate goal of streamflow separation should be to determine the contribution of each component to streamflow, and not merely to determine the contribution of these two generalized components. Therefore, it is more meaningful to determine the contribution of various components by comparing different baseflow separation methods than to use one method to calibrate another.

In fact, some studies have shown that the baseflow sequences generated by the CMB method usually include some high-conductivity water flushed out from swamps or depressions by rainfall, and do not include transient water with low conductivity, such as bank storage return flow and interflow (Cartwright et al., 2014; McCallum et al., 2010; Yang et al., 2019b). However, to date, no research has clearly identified whether these transient water sources are included or excluded in the baseflow sequences generated by the Eckhardt method. Unlike these studies, the present study attempted to resolve this confusion by conducting a detailed analysis of the effect of correcting the Eckhardt method against the CMB method and further analysis of whether the Eckhardt method is truly corrected by the CMB method. The present study not only focused on the consistency of cumulative baseflow or the BFI value, but also on the degree of fit and deviation of the daily baseflow sequence after correction. In addition, the present study discussed in detail the effects of transient water sources on streamflow and conductivity and the different transient water sources included in the separation results of the two methods.

## 2 Methods

### 2.1 Two-parameter recursive digital filter method (Eckhardt)

The filtering method uses the basic principle of baseflow constituting the low frequency component of streamflow that reacts relatively slowly to precipitation, whereas surface runoff constitutes the high frequency component of streamflow that reacts quickly to precipitation (Xie et al., 2020). Eckhardt (2005) combined the basic principles of the filtering method with the linear reservoir model, which reflects the linear relationship between discharge and storage of groundwater in a basin to derive the Eckhardt filter equation (Eq. 1):

$$b_k = \frac{(1-\text{BFI}_{\max})\alpha b_{k-1} + (1-\alpha)\text{BFI}_{\max}y_k}{1-\alpha\text{BFI}_{\max}} \tag{1}$$

Eq. (1) is limited by $b_k \leq y_k$, $\alpha$ is the recession constant, $\text{BFI}_{\max}$ is the maximum baseflow index (the long-term ratio of baseflow to total streamflow), $b_k$ is the baseflow and $y_k$ is the streamflow for the time step $k$.

Eckhardt (2008) proposed a recession analysis method for the calculation of the recession coefficient ($\alpha$). Under conditions of the streamflow recession phase satisfying Eq. (2) and persisting over an extended period, $y_{k+1}$ and $y_k$ can be considered equal to the baseflow. Eq. (3) can then be established if the theoretical assumption of a linear reservoir is true.

$$y_{k-3} > y_{k-2} > y_{k-1} > y_k > y_{k+1} > y_{k+2} \tag{2}$$

$$y_{k+1} = \alpha y_k \tag{3}$$

The slope of the upper boundary of the scatter plot of all $y_{k+1}$ and $y_k$ that meet the above conditions can be considered as $\alpha$, which usually has a random error of less than 2% (Eckhardt (2008).

Eckhardt (2005) suggested the selection of $\text{BFI}_{\max} \approx 0.80$ for perennial streams with porous aquifers, $\text{BFI}_{\max} \approx 0.50$ for ephemeral streams with porous aquifers and $\text{BFI}_{\max} \approx 0.25$ for perennial streams with hard rock aquifers. Collischonn and Fan (2013) proposed a reverse iterative algorithm [Eq. (4)] for estimating $\text{BFI}_{\max}$ based on the linear reservoir assumption. Eq.

(4) is iterated in the reverse direction to obtain the maximum daily baseflow, following which the sum can be divided by the total streamflow to obtain the $BFI_{max}$. The present study used this approach to estimate the $BFI_{max}$ before correction.

$$b_{k-1} = \frac{b_k}{\alpha} \ (b_{k-1} \leq y_{k-1}) \tag{4}$$

## 2.2 Conductivity mass balance method (CMB)

The two-component mass balance method using conductivity as a tracer (CMB) has been proposed by Pinder and Jones (1969) and Yu and Schwartz (1999). Eq. (5) shows the general form of the CMB, which is based on three implicit assumptions: (1) apart from baseflow and surface runoff, the contributions of other flow components can be ignored; 2) the conductivities of surface runoff and baseflow are constant or change in a predicted manner, and show obvious differences during the separation period; 3) in-stream processes such as evaporation do not significantly change the conductivity (Miller et al., 2014; Yang et al., 2019a).

$$b_k = \frac{y_k(SC_k - RO_C)}{BF_C - RO_C} \tag{5}$$

In Eq. (5), $SC_k$, $BF_C$ and $RO_C$ are the conductivities of streamflow, baseflow and rainfall runoff, respectively. Generally, conductivity reflects the total salinity or the concentration of charged ions of streamflow. Some studies have pointed out a positive correlation between conductivity and chloride ion content (Cartwright et al., 2014). A field study by Stewart et al. (2007) showed that the maximum and minimum streamflow conductivities of a basin can be used as an estimate of $BF_C$ and $RO_C$, respectively. However, the maximum conductivity of streamflow may be a function of the combined effects of evaporation, human activities and baseflow, whereas estimation of the minimum conductivity may be affected by instrument errors. Therefore, Miller et al. (2014), Yang et al. (2019a) and Lyu et al. (2020) recommend the use of the conductivity value at 99% probability of each year as an estimate of baseflow conductivity, whereas gaps in the yearly baseflow conductivity timeseries can be obtained by linear interpolation. They also recommended the use of the conductivity value at 1% probability in all records as an estimate of surface runoff conductivity. The present study followed the strategy of Miller et al. (2014). It

should be noted that in some humid regions, the maximum value of streamflow conductivity may be lower than that of regional

groundwater (Cartwright and Irivine, 2020; McCallum et al., 2010). This may be caused by the continuous discharge of

transient water sources into the river during the dry season. Due to the lack of regional groundwater conductivity data, the

present study assumed that this phenomenon was not exist in each basin in which the CMB method was applied.

## 2.3 Calibration of the Eckhardt method

Similar to Lott and Stewart (2016), Zhang et al. (2013), and Saraiva Okello et al. (2018), the present study calibrated the $BFI_{max}$

by minimizing the deviation between the baseflow series separated by CMB and Eckhardt methods. During the correction, the

absolute relative bias (PBIAS) between the daily baseflow series calculated by the Eckhardt and CMB methods was used as

the objective function, and $BFI_{max}$ was gradually adjusted at intervals of 0.01 until a minimum absolute PBIAS was obtained.

## 2.4 Evaluation of the calibration effect

This present study calculated the Nash–Sutcliffe (Nash and Sutcliffe, 1970) efficiency coefficient (NSE) and relative bias

(PBIAS) between the cumulative baseflows obtained by the Eckhardt and CMB methods, and also the NSE, PBIAS, PBIAS(-),

PBIAS(+) and P(|daily bias| >50%) between the daily baseflows (Eqs. 6-10) to evaluate the calibration effect.

$$NSE = 1 - \frac{\sum_{i=1}^{n}(Q_o^i - Q_m^i)^2}{\sum_{i=1}^{n}(Q_o^i - \overline{Q_o^i})^2} \tag{6}$$

$$PBIAS = \frac{\sum_{i=1}^{n}(Q_m^i - Q_o^i)}{\sum_{i=1}^{n} Q_o^i} \times 100\% \tag{7}$$

$$PBIAS(-) = \frac{\sum_{i=1}^{n}(Q_m^i - Q_o^i)}{\sum_{i=1}^{n} Q_o^i} \times 100\% \quad (Q_m^i < Q_o^i) \tag{8}$$

$$PBIAS(+) = \frac{\sum_{i=1}^{n}(Q_m^i - Q_o^i)}{\sum_{i=1}^{n} Q_o^i} \times 100\% \quad (Q_m^i > Q_o^i) \tag{9}$$

$$P(|daily\ bias| > 50\%) = \frac{The\ number\ of\ \left|\frac{b_{k(ECK)} - b_{k(CMB)}}{b_{k(CMB)}}\right| > 0.5}{The\ number\ of\ total\ time\ steps} \times 100\% \tag{10}$$

In Eq. (6) to Eq. (10), $Q_o$ is the reference standard value or observation value, $Q_m$ is the simulation or calculation value, the NSE reflects the degree of fit of the two series, the PBIAS reflects the total relative deviation between the two series, PBIAS(-) reflects the total negative relative deviation between two series, PBIAS(+) reflects the total positive relative deviation between two series and P(|daily bias| >50%) reflects the proportion of the sequences with > 50% absolute daily bias between the two series. The closer the NSE is to 1, the better the fit between the simulated and observed values, whereas closer the PBIAS value is to 0, the smaller the deviation between the simulated and observed values and the larger the value of P(|daily bias| >50%), the greater the proportion of the sequences with obvious deviations between the two baseflow series.

**3 Data**

Lyu et al. (2020) showed that a negative correlation between streamflow and conductivity is an indicator of the applicability of the CMB method, and emphasized that the CMB method has better applicability when the correlation coefficient is less than −0.5. The adoption of this criterion also eliminates those basins which are clearly affected by human activities such as reservoirs and sewage discharge, since the impact of human activities will reduce the negative correlation between streamflow and conductivity(Miller et al., 2014). Since the estimation of the parameters ($BF_C$, $RO_C$) of the CMB method may have greater uncertainty when the time series is short, Lyu et al. (2020) suggested that the time series should exceed 6 months whereas Lott and Stewart (2016) suggested that the time series should exceed 2 years. Therefore, the present study randomly selected 26 hydrological stations from the United States Geological Survey (USGS) National Water Information System (NWIS) website: http://waterdata.usgs.gov/nwis (last accessed: September 2020). The negative correlation between conductivity and streamflow at each site was less than −0.5 and the sequence length of each site exceeded 2 years. The areas of the basins gauged by these hydrological stations range from 46 $km^2$ to 110,973 $km^2$ and the lengths of the measured streamflow timeseries among the stations range from 2 to 9 years. The streamflow data of all stations are continuous and complete, and no station has missing conductivity data exceeding 10% of total data. The present study performed linear interpolation based on the conductivity values at both ends of the missing period to infill missing data. Table S1 shows a summary of the hydrological stations used. The present study used the Eckhardt and CMB methods to separate the baseflow of these 26 stations, following

which the $BFI_{max}$ of the Eckhardt method in each station was calibrated with reference to the CMB method. Finally, the effect of the correction was evaluated and discussed.

## 4 Results

Table S1 shows the results for the estimation of the parameters ($\alpha$, $BFI_{max}$). The $\alpha$ values of the 26 stations ranged from 0.978 to 0.998 with an average of 0.991. Before calibration, the $BFI_{max}$ ranged from 0.19 to 0.86 with an average of 0.39, whereas after correction, the $BFI_{max}$ ranged from 0.17 to 0.67 with an average of 0.39. Although the average value of $BFI_{max}$ did not change after calibration, the range of fluctuation was reduced.

Table 1 shows the baseflow separation results of the CMB and Eckhardt methods before and after correction. The baseflow index ($BFI = \frac{\sum b_k}{\sum y_k}$) calculated by the CMB method was between 0.15 and 0.64 with an average of 0.29. The BFI calculated by the Eckhardt method before calibration was between 0.14 and 0.81 with an average of 0.31, whereas that after calibration was between 0.15 and 0.63 with an average of 0.29. As shown in Fig. 1, there was an element of random deviation between the BFI values calculated by the Eckhardt and CMB methods before calibration, with that of the Eckhardt method showing no obvious trend of overestimation or underestimation, whereas the BFI values calculated by the two methods were basically identical after calibration.

Table 1 shows the NSE and PBIAS values for the comparison of the cumulative baseflow series by the Eckhardt and CMB methods after calibration. The NSE ranged from 0.91 to 1.00 with an average value of 0.97, whereas the PBIAS ranged from −12% to 13% with an average of −1%. The cumulative baseflow obtained by the Eckhardt method after calibration showed a good fit with that of the CMB method, indicating that the two methods generated consistent estimates of total baseflow after calibration.

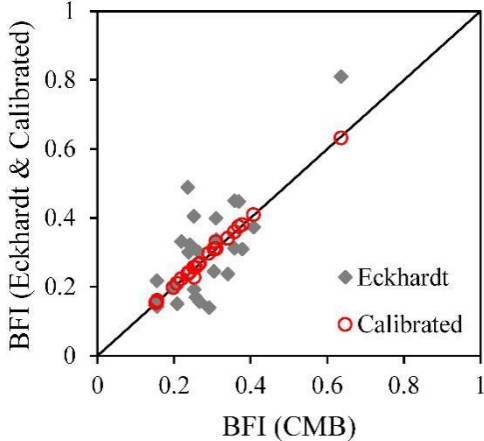

**Figure 1. Comparison of BFI of the Eckhardt and conductivity mass balance (CMB) methods before and after calibration.**

Table 1 also shows the NSE, PBIAS, PBIAS(-), PBIAS(+) and P(|daily bias| >50%) obtained through the comparison of daily

baseflow series generated by the corrected Eckhardt and CMB methods. The NSEs of daily baseflow ranged from −2.35 to

0.45 with an average of −0.30. The NSEs of 20 of the 26 stations were less than zero, indicating that the daily baseflow series

generated by the two methods showed major differences. The PBIAS ranged from −11% to 2% with an average of 0%, the

PBIAS(-) ranged from −47% to −13% with an average of −28% and the PBIAS(+) ranged from 14% to 48% with an average

of 28%. Fig. 2 shows the variations in PBIAS, PBIAS(-) and PBIAS(+) with $BFI_{max}$ using station 02297100 as an example,

where it is evident that an increase in $BFI_{max}$ results in a gradual increase in total relative deviation (PBIAS) from negative to

positive. The total deviation (PBIAS) is zero when the absolute values of PBIAS(-) and PBIAS(+) are equal and offset each

other. In other words, although the two methods obtain the same total baseflow after calibration, some obvious biases between

the daily baseflow series persist, although the positive and negative biases offset each other. The P (|daily bias| >50%) ranged

from 13% to 84% with an average of 44%. On average, nearly half of the daily baseflow series obtained by the two methods

after calibration showed a daily absolute bias exceeding 50%. Therefore, it can be argued that the calibration of the Eckhardt

method against the CMB method obtains a "false" correction under which the same total baseflow series obtained by the two

methods was due to the offsetting of inherent deviation in the baseflow series.

**Table 1.** The results of baseflow separation of streamflow from 26 hydrological stations of the United States Geological Survey (USGS) National Water Information System (NWIS) before and after calibration. "Eck." represents the uncalibrated Eckhardt method whereas "Cali." represents the calibrated Eckhardt method.

| Sit number | BFI | | | Accumulated baseflow | | Daily baseflow | | | | P (\|daily bias\|[#] >50%) |
|---|---|---|---|---|---|---|---|---|---|---|
| | CMB | Eck. | Cali. | NSE | PBIAS | NSE | PBIAS | PBIAS (-) | PBIAS (+) | |
| 02298202 | 0.31 | 0.34 | 0.31 | 0.99 | 4% | -0.75 | -1% | -39% | 38% | 54% |
| 02303000 | 0.34 | 0.24 | 0.34 | 0.95 | -11% | -0.75 | 0% | -23% | 23% | 68% |
| 02306774 | 0.16 | 0.14 | 0.16 | 0.96 | -3% | -0.11 | 1% | -47% | 48% | 84% |
| 02297100 | 0.25 | 0.19 | 0.23 | 0.98 | 0% | -0.55 | -11% | -39% | 28% | 44% |
| 08068275 | 0.15 | 0.15 | 0.15 | 0.93 | 13% | 0.11 | 0% | -37% | 36% | 51% |
| 02160105 | 0.36 | 0.45 | 0.36 | 0.97 | -10% | 0.09 | 0% | -17% | 17% | 40% |
| 02160700 | 0.37 | 0.45 | 0.38 | 0.97 | -9% | 0.06 | 1% | -17% | 18% | 39% |
| 02207120 | 0.24 | 0.30 | 0.24 | 0.99 | 5% | 0.05 | 0% | -19% | 20% | 42% |
| 03007800 | 0.22 | 0.33 | 0.22 | 0.99 | 3% | -0.10 | 1% | -30% | 31% | 46% |
| 03044000 | 0.26 | 0.17 | 0.26 | 0.96 | -7% | -0.40 | -1% | -27% | 27% | 41% |
| 03072655 | 0.26 | 0.30 | 0.27 | 0.99 | 5% | -0.07 | 0% | -32% | 32% | 51% |
| 03106000 | 0.20 | 0.20 | 0.20 | 0.91 | -12% | -0.07 | -1% | -35% | 34% | 51% |
| 03201980 | 0.27 | 0.16 | 0.27 | 0.99 | 7% | 0.02 | -1% | -36% | 35% | 50% |
| 03321500 | 0.30 | 0.25 | 0.31 | 0.98 | 7% | -0.08 | 1% | -25% | 26% | 40% |
| 06037500 | 0.64 | 0.81 | 0.63 | 0.97 | -8% | -1.06 | 0% | -13% | 14% | 13% |
| 06296120 | 0.41 | 0.37 | 0.41 | 0.95 | -10% | -0.42 | 0% | -26% | 26% | 45% |
| 06711565 | 0.16 | 0.22 | 0.16 | 0.97 | -1% | -0.63 | -1% | -38% | 38% | 44% |
| 07079300 | 0.25 | 0.41 | 0.26 | 0.99 | -4% | -0.25 | 1% | -36% | 37% | 59% |
| 07086000 | 0.24 | 0.32 | 0.24 | 1.00 | 0% | -0.08 | 0% | -18% | 18% | 22% |
| 07119700 | 0.38 | 0.31 | 0.38 | 0.98 | 6% | -0.32 | 0% | -28% | 28% | 38% |
| 03036000 | 0.29 | 0.14 | 0.30 | 1.00 | 0% | -0.09 | 1% | -24% | 26% | 41% |
| 03067510 | 0.21 | 0.15 | 0.21 | 0.97 | -8% | -0.17 | 0% | -28% | 28% | 43% |
| 03374100 | 0.31 | 0.40 | 0.33 | 1.00 | 3% | 0.45 | 2% | -18% | 20% | 36% |
| 06089000 | 0.36 | 0.31 | 0.36 | 0.99 | 3% | -0.08 | 0% | -19% | 19% | 21% |
| 07081200 | 0.24 | 0.49 | 0.24 | 0.95 | -12% | -2.35 | -1% | -31% | 30% | 55% |
| 07097000 | 0.42 | 0.37 | 0.42 | 0.99 | 5% | -0.20 | 1% | -18% | 19% | 16% |
| Average | 0.29 | 0.31 | 0.29 | 0.97 | -1% | -0.30 | 0% | -28% | 28% | 44% |
| SD | 0.10 | 0.14 | 0.10 | 0.02 | 7% | 0.52 | 2% | 9% | 8% | 15% |

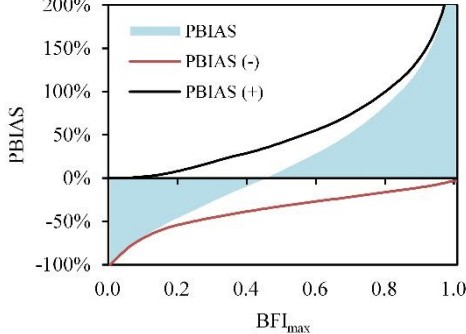

**Figure 2. The biases between daily baseflow series calculated by the Eckhardt and CMB methods for the United States Geological Survey (USGS) station 02297100 varied with BFI$_{max}$. PBIAS is total relative bias, PBIAS(-) is the total negative relative bias and PBIAS(+) is the total positive relative bias.**

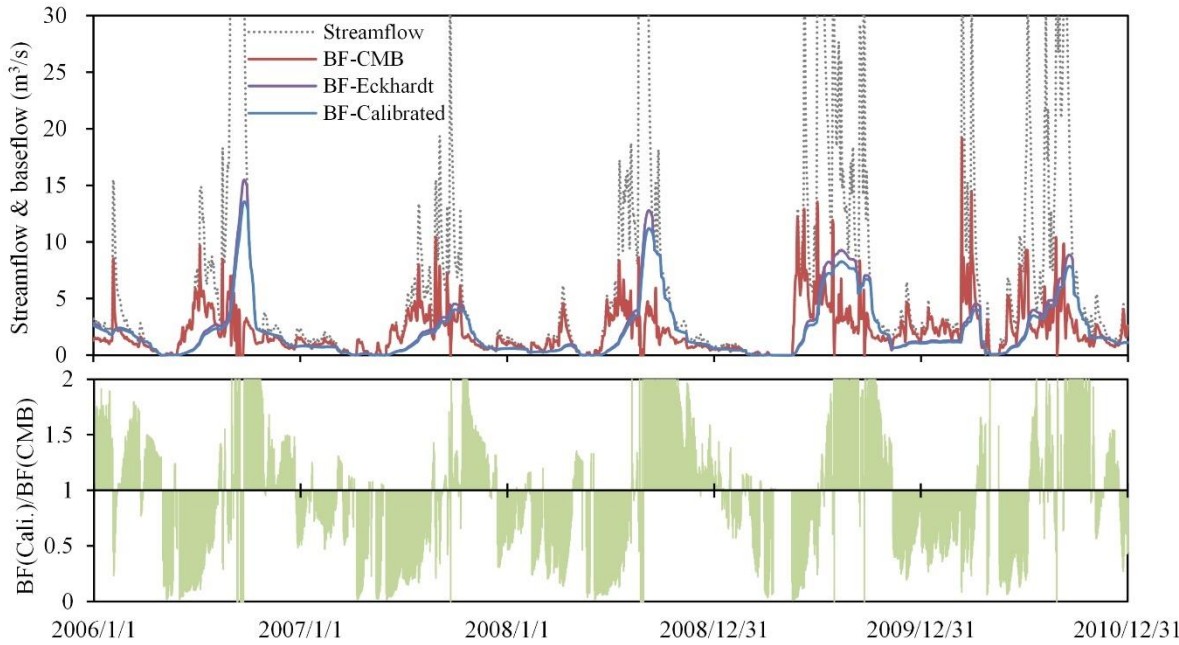

**Figure 3. Bias between daily baseflow series generated by the conductivity mass balance (CMB) and Eckhardt methods after calibration for the United States Geological Survey (USGS) station 02298202. BF represents baseflow. The lines beyond the coordinate range are not shown to show the details more clearly.**

Figure 3 shows the bias between the daily baseflow series generated by the CMB and Eckhardt methods after calibration using station 02298202 as an example. As shown in Figure 3, the peak of the baseflow sequence generated through the Eckhardt method usually appeared during the recession stage, whereas that through the CMB method usually appeared during the rising stage. This misaligned peak resulted in obvious periodicity in deviation between the baseflow sequences obtained by the Eckhardt and CMB methods. The baseflow generated by the Eckhardt method was usually significantly lower than that generated by the CMB method during the rising stage, whereas it was significantly higher during the recession stage. This phenomenon was also reflected in other stations, as shown for an additional five stations in Supplement 1 (Fig. S1–S5). The examples mentioned above focused on periodic deviation in the rising and falling limbs during the wet season, and the corresponding deviation during the dry season tended to be smaller (Fig. 3, Fig. S1-S5). As discussed by Cartwright et al. (2014), this phenomenon may be due a higher contribution of transient water sources to streamflow during the wet season. In

addition, the calibration process may aggravate the deviation observed during the dry season, with Fig. S5 being an obvious example. Before calibration, the two methods obtained basically the same separation results for the dry season, whereas after calibration, although the total deviation was minimized, there was a significant increase in the deviation during the dry season, which is logically incorrect.

## 5 Discussion

### 5.1 The influence of transient water on streamflow and conductivity

As mentioned in the introduction, streamflow includes not only surface runoff and baseflow, but also a variety of different transient water sources, such as interflow, bank storage return flow and high conductivity water flushed out from depressions or wetlands by rainfall (Cartwright et al., 2014; Schwartz, 2007; McCallum et al., 2010; Lin et al., 2007). These transient water sources flow into the river at different spatiotemporal resolutions, thereby affecting streamflow and conductivity. Figure 4 is a conceptual diagram of the influence of different transient water sources on streamflow and conductivity during the late dry season, rainfall and post-rain recession periods.

Ideally, streamflow is mainly dominated by groundwater discharge after a long period of drought during which the conductivity of streamflow in the basin is close to that of groundwater (Lott and Stewart, 2013; Stewart et al., 2007).The conductivity of streamflow needs to be compared with that of regional groundwater to determine whether or when this state is reached in a basin. In addition, continual evaporation will result in a gradual increase in the TDS and conductivity of wetland water, depression water and shallow groundwater in the valley (Liu et al., 2019).

During a rainfall event, a portion of rainfall forms surface runoff, a portion infiltrates the soil to recharge groundwater and a portion of the infiltrated water returns to the surface runoff to form interflow (Nathan and McMahon, 1990; Nejadhashemi et al., 2007; Tallaksen, 1995). Surface runoff formed during the early stage of rainfall will flush out high-conductivity wetland or depression water in the valley, or quickly mobilize high-conductivity soil water, forming a pulse of high-conductivity water (Cartwright et al., 2014; Yang et al., 2019b). This phenomenon is essentially part of the "old" water paradox: the early flood

events in most basins usually contain a lot of "old" water with chemistry or isotopic characteristics different from precipitation (Kirchner, 2003), and there remains no accepted mechanism to explain this phenomenon (Bishop et al., 2004; Kirchner, 2003). However, the flushing of wetlands or depression water and the rapid mobilization of soil water are the two most likely mechanisms to explain this phenomenon (Cartwright and Morgenstern, 2018; Kienzler and Naef, 2008; Xiao et al., 2020). This pulse of high-conductivity water can lead to an overestimation of the conductivity of streamflow at the rising stage, and even the estimation of an abnormal increase in conductivity with an increase in streamflow (Aubert et al., 2013; Cartwright et al., 2014; Zhi et al., 2019). This abnormal increase in conductivity with increasing streamflow can be easily screened out from the conductivity sequences. Figure 5 shows part of the screening results for two stations (06296120 and 03201980) where it is evident that the point of abnormal increase in conductivity is usually distributed during the initial rising stage, and usually corresponds to the peak in baseflow of the CMB method. Continuous rainfall will subsequently result in the flow of a large amount of low-conductivity water into the river, resulting in a significant decrease in conductivity of streamflow approaching that of rainfall. At the same time, the rapid rise in the river water level will result in recharge of the aquifer by part of the low-conductivity streamflow to form bank storage water (Howcroft et al., 2019; McCallum et al., 2010).

During the recession stage after rainfall, surface runoff quickly recedes and stops, whereas interflow gradually decreases and finally stops. The proportion of groundwater in streamflow gradually increases, resulting in a gradual rise in conductivity. At the same time, the low-conductivity bank storage water formed during the rainy season is also gradually returned to the stream (Cartwright et al., 2014; McCallum et al., 2010). There have been many studies on the influence of bank storage and return flow on streamflow and solutes (Cartwright and Irivine, 2020; Chen and Chen, 2003; McCallum et al., 2010; McCallum and Shanafield, 2016). The general consensus among these studies is that low-conductivity river water generated during the flood stage will seep into the aquifer under the action of the hydraulic gradient, and will continue to be discharged for several months after the flood, eventually leading to a significantly delayed solute discharge process. Cartwright et al. (2014) emphasized that interflow is influenced by the same mechanism as that of bank storage return flow. Both interflow and bank storage return flow result in the conductivity of streamflow during the recession stage being lower than that during the rising stage, thereby

forming a clockwise hysteresis loop between conductivity and streamflow. The existence of this hysteresis loop between solute and streamflow has been confirmed by many other studies (Aubert et al., 2013; Evans and Davies, 1998; Wagner et al., 2019; Winnick et al., 2017; Zhi et al., 2019). As shown in Figure 6, this hysteresis loop was evident in all 26 stations examined in the present study. There were usually differences in the shapes of the hysteresis loops among the different stations or different flood events for the same station, which reflects the different effects of bank storage return flow or interflow in different watersheds or in the same watershed at different periods. Cartwright and Miller (2021) analyzed the temporal and spatial variations in river conductivity in Australia (the Barwon, Glenelg, and Campaspe rivers) and North American (the upper Colorado River) basins, and similarly attributed these variations to the temporal and spatial fluctuations of transient water sources.

In addition to these transient water sources, human impacts such as reservoirs, abstraction of water for irrigation and sewage discharge will also affect streamflow and conductivity. Reservoirs have a direct influence on streamflow and solute transport processes (Lehner et al., 2011) and abstraction of river water for irrigation disturbs streamflow. On the other hand, irrigation return flow water tends to have a higher TDS, which results in an increase in the conductivity of streamflow (Kronholm and Capel, 2015). Domestic sewage usually contains a large amount of inorganic salts, and the discharge of sewage can result in a significant increase in the conductivity of streamflow (Osode and Okoh, 2009). Although different basins may be affected by different human activities to different degrees, it is clear that human activities significantly reduce the negative correlation between streamflow and conductivity. The negative correlation coefficients between streamflow and conductivity of the basins examined in the present study were all less than -0.5, therefore, basins that were obviously affected by human activities have been excluded. For this reason, the present study only briefly discussed the impact of human activities.

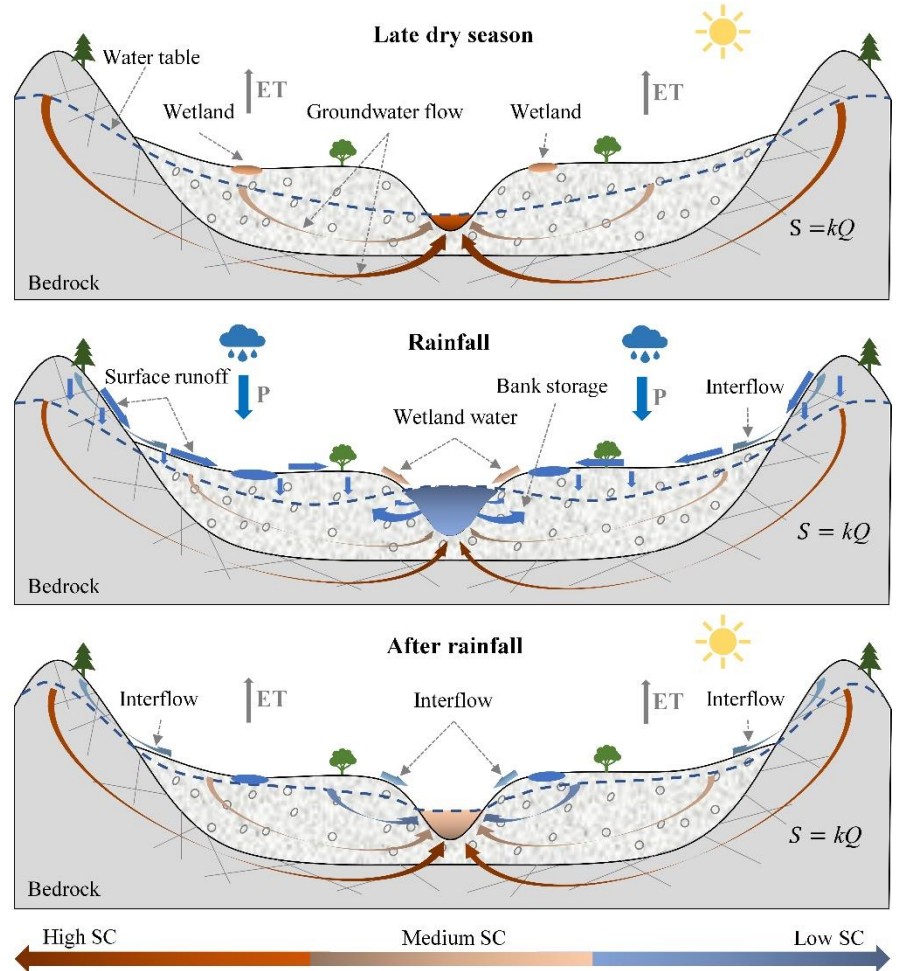

**Figure 4. A conceptual diagram illustrating the influence of different transient water sources on streamflow over different periods. For the streamline arrows in the figures, color reflects the relative conductivity whereas width reflects the relative flow. The**
320 **equation shown represents the linear reservoir model describing the relationship between groundwater storage (*S*) and discharge (*Q*).**

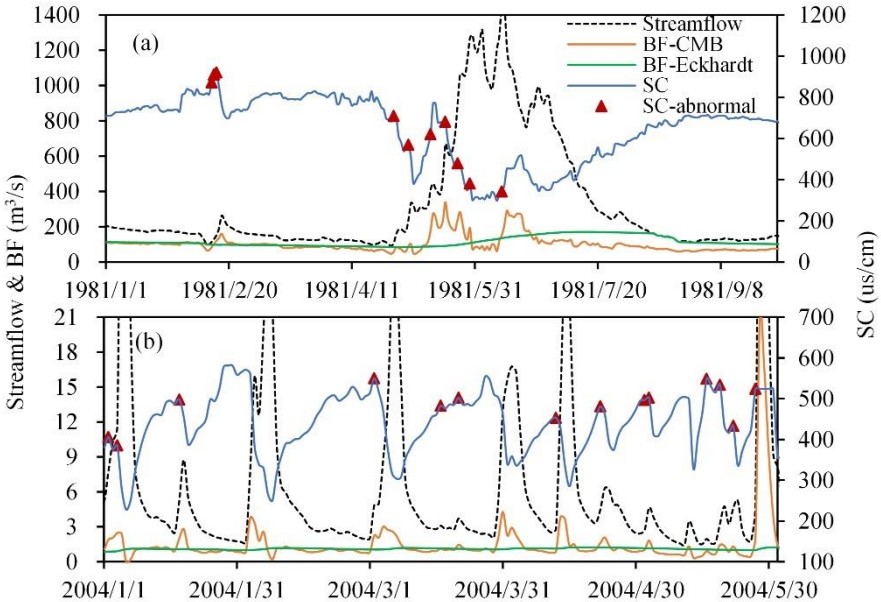

**Figure 5. The abnormal increase in conductivity resulting from the flushing out of concentrated high-conductivity water from wetlands or depressions during the initial rising stage evident in the streamflow sequences of two United States Geological Survey (USGS) stations. (a) station 06296120, (b) station 03201980. It was assumed that this abnormal increase in conductivity was present when an increase in streamflow exceeding 10% was accompanied by an increase in conductivity. BF: baseflow, SC: conductivity.**

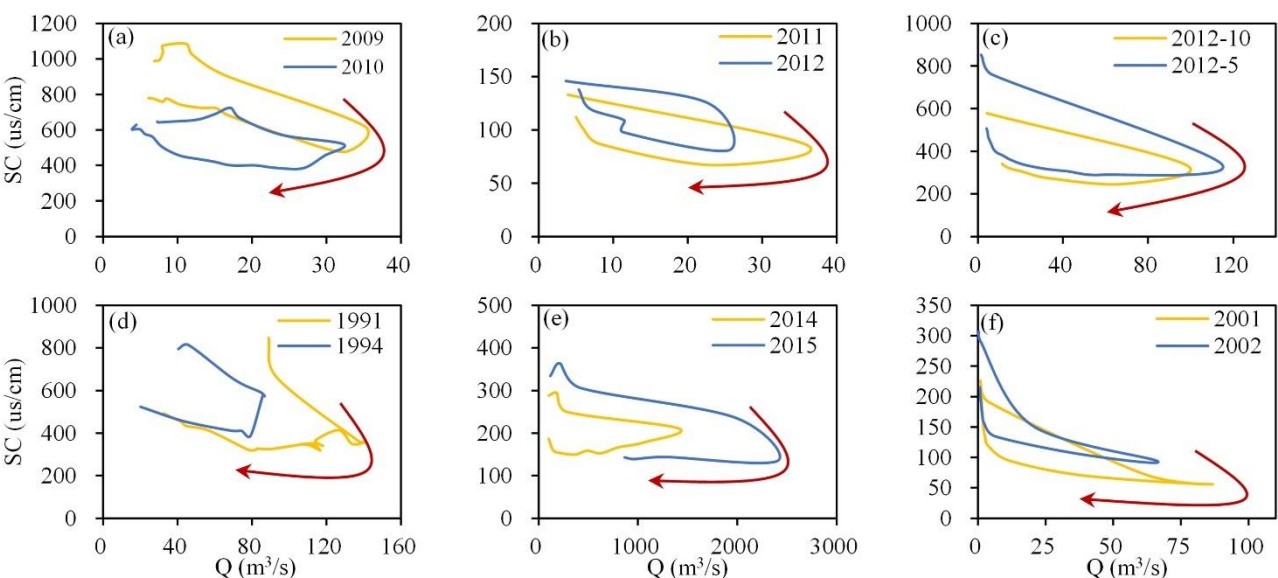

**Figure 6. Clockwise hysteresis loops between conductivity and streamflow during flood events evident in the streamflow sequences of six United States Geological Survey (USGS) stations. (a) station 02298202; (b) station 02207120; (c) station 03106000; (d) station 06089000; (e) station 03072655; (f) station 08068275. The arrows reflect the direction of time.**

## 5.2 The transient water components are different among the baseflow sequences generated by the CMB and Eckhardt methods

The results of the present study (Section 4) confirmed that it is not possible to calibrate the Eckhardt method against the CMB method as the baseflow series generated by these two methods show inherent deviations. These inherent deviations are mainly due to the baseflow series generated by the two methods containing different transient water sources (Cartwright et al., 2014; Hagedorn, 2020; Rammal et al., 2018) as the two methods are constructed based on different theoretical assumptions (Section 2).

The Eckhardt method subscribes to the linear reservoir model ($S = kQ$) between discharge ($Q$) and storage ($S$) of groundwater in a basin, where $k$ is the recession constant and its relationship with the filtering parameter $\alpha$ is: $k = \frac{-1}{\ln(\alpha)}$) (Chapman, 1999). The linear reservoir model can be derived based on the Boussinesq equation and Darcy's law of porous media (Brutsaert and Nieber, 1977; Furey and Gupta, 2000). Many studies based on recession analysis have confirmed the universal existence of this linear reservoir relationship (Brutsaert, 2008; Tallaksen, 1995; Thomas et al., 2013). The baseflow sequence obtained based on the linear reservoir theoretical assumption usually has the following characteristics (Duncan, 2019): (1) the recession of baseflow will continue for an extended period after the rise of streamflow; (2) the baseflow peak usually appears after the streamflow peak due to the storage-routing effect of underground reservoirs; (3) baseflow recession is likely to follow an exponential decay function, i.e., the linear reservoir model. Therefore, the baseflow sequence separated by the Eckhardt method theoretically does not generally include high-conductivity water flushed out from wetlands or depressions by rainfall at the beginning of the rising stage, but does include the majority of water flowing through the porous medium to satisfy the linear reservoir assumption, including groundwater flow, slow interflow and bank storage return flow.

The CMB method subscribes to a chemical mass balance under which separated baseflow usually comprises components with high conductivity, regardless of whether these components flow through a porous medium or whether they meet the linear reservoir assumption. Therefore, the baseflow sequence generated by the CMB method will include high-conductivity water flushed out of wetlands or depressions by rainfall, but will not include interflow and bank storage return flow with low conductivity (Cartwright et al., 2014; Rammal et al., 2018). The flushing out of high-conductivity water from wetlands or depressions mainly occurs during the initial rising stage, while interflow and bank storage return flow mainly occur during the recession stage. Therefore, the baseflow sequences generated by Eckhardt and CMB methods include different transient water sources and show obvious misaligned peaks and periodic deviation during the wet season (Fig. 3, Fig. S1-S5). Although the deviation between the two methods is relatively small during the dry season, some inherent deviation between them may persist. The Eckhardt method usually recognizes many segments as fully baseflow segments when they meet the recession characteristics, but the CMB method only recognizes a full baseflow segment when the conductivity reaches the maximum for 1 or 2 days. The $BFI_{max}$ calibration process will only change the total baseflow, and will not significantly change the position of the baseflow peak and the frequency of the full baseflow segment. Therefore, the use of the CMB as a reference to calibrate the Eckhardt method is not recommended, even if it is only carried out during the dry season.

Given that the results of baseflow separation by the two methods show inherent deviations, future research should avoid investing a lot of time on using one method to calibrate the other and instead focus on analyzing the underlying causes of these inherent deviations and their significance for the study of hydrological processes. For example, by comparing the baseflow sequences calculated by these two methods, streamflow can be separated into multiple components or the contribution of different transient water sources to streamflow can be identified. Figure 7 is a schematic of different streamflow components reflected by inherent deviations between the two baseflow sequences. The intersection of the baseflow sequences of the two methods reflects high-conductivity deep circulating groundwater with a long residence time. The portion of the baseflow sequence generated by the Eckhardt method that is situated above that generated by the CMB method reflects low-conductivity transient water with a short residence time such as bank storage return flow and interflow. The portion of the baseflow sequence

generated by the CMB situated above that by the Eckhardt method reflects high-conductivity transient water such as high-conductivity water from wetlands or depressions. The complementary set of the two baseflow separation results reflects surface runoff.

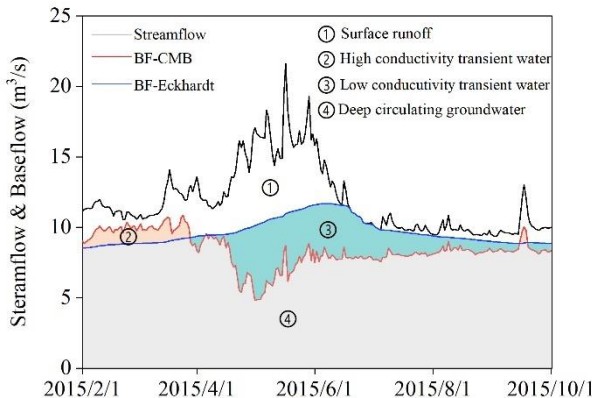

**Figure 7. Schematic diagram showing the inherent deviation between baseflow sequences generated by the Eckhardt and**
**conductivity mass balance (CMB) methods which reflects different streamflow components using the United States Geological Survey (USGS) station 06037500 as an example. BF: baseflow.**

**6 Conclusions**

The present study evaluated the effectiveness of calibrating the Eckhardt method against the CMB method for 26 basins in the United States by comparing biases between the daily baseflow sequences generated by the two methods and attempted to
resolve the confusion resulting from the combined application of the two methods from a new perspective. Compared with previous studies, the basins examined in the present study cover a wider range of climate and basin characteristics. The areas of the basins studied ranged between 46 km² to 110,973 km², with the latitudes of the hydrological stations ranging between 26.98°N to 47.52°N. Therefore, the basins examined in the present study were representative of most climate zones from Florida to Montana. The main conclusions drawn from the present study are summarized below.

The calibration of the Eckhardt method against the CMB method represents a "false" correction based on only the total

baseflow by offsetting inherent biases in the baseflow sequences generated by the two methods. The use of the two-component

CMB method as a reference to calibrate the Eckhardt method is not recommended, even if it is only performed during the dry

season.

The present study verifies and extends the conclusions obtained in previous research (McCallum et al., 2010; Cartwright et al.,

2014) under a wider range of watershed characteristics. These conclusions include that different transient water sources may

contribute to baseflow separated by methods based on different basic assumptions. The baseflow sequence generated by the

Eckhardt method usually includes slow interflow and bank storage return flow, whereas that of the CMB method usually

includes high-conductivity water flushed from swamps and depressions by rainfall, but excludes low-conductivity interflow

and bank storage return flow. The differences in transient water sources contributing to baseflow between these two methods

results in differences in the baseflow sequences obtained, particularly in terms of an obvious misalignment of flow peaks and

periodic deviation. Therefore, the one method cannot be used to calibrate the other (Fig. 3). However, the four-component

separation of streamflow can be achieved through comparison (Fig. 7).

These results of the present study can also provide some hydrological insight: a) The application of two-component baseflow

separation methods based on different theoretical assumptions is likely produce baseflow series containing different

components. In application, there should be careful analysis of whether the components contained in the separation results are

consistent with the research objectives. b) The adoption of the definition of baseflow as the amount of discharge of regional

groundwater to a river (which is used to evaluate surface water and groundwater interactions) within the CMB and Eckhardt

methods can result in large errors in the rising and falling limbs, respectively. Therefore, the use of the Eckhardt method and

CMB method for the rising and falling limbs, respectively, may be more reasonable. c) Future research should consider the

existence of deviations between the separation results produced by different baseflow separation methods, try and identify the

reasons for these differences, and explore the hydrological information contained therein. In addition, it should be recognized

that mutual correction cannot be used to obtain consistent daily results between different baseflow separation methods.

**Data availability:** All streamflow and conductivity data can be retrieved from the US Geological Survey's (USGS) National Water Information System (NWIS) website using the special gage number: http://waterdata.usgs.gov/nwis (NWIS, 2020).

**Supplement:** The supplement related to this article is available online at: Supplement 1.

**Author contributions:** WY, CX, ZZ, and XL developed the research train of thought. WY and ZZ collected and processed all research data. CL and XL applied for funding for this research. WY prepared the manuscript with contributions from all coauthors.

**Competing interests:** The authors declare that they have no conflict of interest.

**Acknowledgements:** This work is supported by the National Natural Science Foundation of China (41572216), the Provincial School Co-construction Project Special – Leading Technology Guide (SXGJQY2017-6), the China Geological Survey Shenyang Geological Survey Center "Hydrogeological Survey of Songnen Plain" project ([2019]DD20190340-W09), and the Geological Survey Foundation of Jilin Province (2018-13). We thank the anonymous reviewers for useful comments to improve the manuscript.

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
