# Peer review of "Can the two-parameter recursive digital filter baseflow separation method really be calibrated by the conductivity mass balance method?"

_Hydrology and Earth System Sciences, 2020_

## Referee Comment (RC1) · Anonymous Referee #1 · 24 Nov 2020

The ECK method and CMB method are two widely-used baseflow separation methods. The ECK method only requires the stream discharge data as input, which is one of the most readily available methods for baseflow separation in longterm studies. However, the parameters for the ECK method are often subjectively determined, resulting in high uncertainties in the baseflow separation estimations. On the other hand, the CMB method is considered to be more objective because it is based on the direct measurements of streamflow conductivity. However, the data required for the CMB method may not be available for long periods. Using the baseflow data estimated with the CMB method to calibrate parameters for the ECK model can be a more accurate baseflow separation method. The manuscript compared the differences between

two baseflow separation methods (the conductivity mass balance method (CMB) and the two parameters filtering method (Eckhardt) constructed based on different theoretical assumptions. The manuscript examines the correction effect of the CMB method on the Eckhardt method by analyzing the deviation between the daily baseflow series. In addition, the effects of transient water sources on streamflow, conductivity, and baseflow separation results were discussed in detail. They attributed the difference between the two baseflow separation results to the inclusion of different transient water sources, which will provide future researchers with a reference when using and comparing different baseflow separation methods. In my opinion, there is no problem with the overall structure and content of this manuscript, and it can be published after some minor revisions: 1. The author mention many times that "surface runoff formed during the early stage of rainfall will flush out high-salinity wetland or depression water in the valley, forming a high-salinity pulse". However, not every region has such topographical conditions, but the rapid increase of base flow is a very common phenomenon, so more conditions should be discussed. 2. Lines 104-106: "Section 2 introduces these. . . . . .conclusions". This sentence is not necessary here, so suggest to rewrite or delete it. 3. Lines 287-289: "These human activities. . . . . .present study". As you discussed, human activities (reservoir construction, irrigation, sewage discharge) could disturb streamflow and conductivity. In my opinion, these activities will obviously change the original negative power function relationship between conductivity and streamflow. Therefore, it is possible to determine whether it is affected by human activities by analyzing the correlation between conductivity and streamflow. In fact, you have explained in Lines 173-174 that the negative correlation between streamflow and conductivity of the basins used in this study is less than -0.5, in other words, you have excluded those basins that are obviously affected by human activities. Therefore, I suggest to rewrite the sentence of Lines 287-289 to clearly explain the impact of human activities. 4. Lines 332-336: "high-salinity deep circulating groundwater", "low-salinity groundwater", "high-salinity surface water". What is the relationship between salinity and conductivity? There is no clear explanation in the article, which

may confuse readers. Therefore, I suggest that the "salinity" in the text should be replaced by "conductivity", including the abstract, Figure 7, and conclusion. Or explain the relationship between conductivity and salinity in detail at an appropriate place. 5. The Figures should be replaced by more clearer pictures. 6. There are some problems in spelling, grammar, expression and format.

---

## Author Comment (AC1) · 14 Dec 2020

**Response to reviewer #1**

*Explanation: The reviewer' comments are shown in black, the author's replies and revises are shown in blue. The quoted line numbers and page numbers are the Marked manuscripts.*

**Comments from Referee #1:**

The ECK method and CMB method are two widely-used baseflow separation methods. The ECK method only requires the stream discharge data as input, which is one of the most readily available methods for baseflow separation in long term studies. However, the parameters for the ECK method are often subjectively determined, resulting in high uncertainties in the baseflow separation estimations. On the other hand, the CMB method is considered to be more objective because it is based on the direct measurements of streamflow conductivity. However, the data required for the CMB method may not be available for long periods. Using the baseflow data estimated with the CMB method to calibrate parameters for the ECK model can be a more accurate baseflow separation method.

**Response:** Thank you very much for reviewing our manuscript. Indeed, as you said, the CMB method is generally regarded as a relatively objective baseflow separation method. Many studies used the CMB method as a reference to correct the parameters of the Eckhardt method. We also discussed related research in detail in the manuscript **(Page 4, Lines: 75-84)**. However, some studies have pointed out that the separation results of the two methods may contain different transient water sources. These transient water sources may cause inherent deviations in the two baseflow separation results and cannot be truly corrected **(Pages 4-5, Lines: 84-95)**. These two types of views seem to be opposed and may cause confusion for researchers. The main purpose of this research is to try to solve this confusion by analyzing the correction effect of the CMB method on the Eckhardt method **(Page 5, Lines: 96-106)**.

The manuscript compared the differences between two baseflow separation methods (the conductivity mass balance method (CMB) and the two parameters filtering method (Eckhardt) constructed based on different theoretical assumptions. The manuscript examines the correction effect of the CMB method on the Eckhardt method by analyzing the deviation between the daily baseflow series. In addition, the effects of transient water sources on streamflow, conductivity, and baseflow separation results were discussed in detail. They attributed the difference between the two baseflow separation results to the inclusion of different transient water sources, which will provide

future researchers with a reference when using and comparing different baseflow separation methods. In my opinion, there is no problem with the overall structure and content of this manuscript, and it can be published after some minor revisions:

**Response:** Thank you very much for your appreciation of our research. We have revised the manuscript based on your comments. See the following text for one by one reply.

1. The author mention many times that "surface runoff formed during the early stage of rainfall will flush out high-salinity wetland or depression water in the valley, forming a high-salinity pulse". However, not every region has such topographical conditions, but the rapid increase of baseflow is a very common phenomenon, so more conditions should be discussed.

**Response:** Indeed, we cannot be sure that there are wetlands or depressions water in every basin. However, many studies have pointed out the rapid increase of "baseflow" in flood events. The quotes are added because this phenomenon is part of an unresolved paradox: the "old" water paradox. The "old" water paradox refers to the phenomenon that a large amount of old water with water chemical or isotopic characteristics different from precipitation is contained in the streamflow in the early stage of flood event (Kirchner, 2003). Kirchner (2003) pointed out that the increase in hydraulic head gradient and baseflow fluxes caused by precipitation infiltration cannot explain the rapid mobilization of old water, that is, the old water is likely to come from other water sources besides precipitation and baseflow. Some researchers try to explain this paradox based on different theories, such as propagation of pressure waves, transmissivity feedback, soil water mobilization and wetlands water flushing, but no one theory has been universally recognized (Bishop et al., 2004; Kienzler and Naef, 2008; Kirchner, 2003; Cartwright and Morgenstern, 2018). In our opinion, a common understanding clarified by these studies is that the formation of old water is due to the rapid mobilization of a large amount of soil water, depressions or wetlands water by surface runoff, rather than an increase in baseflow. Considering that the original discussion may be relatively simple, we have added some discussion about the "old" water paradox, see **Pages:13-14, Lines: 250-256**.

2. Lines 104-106: "Section 2 introduces these*: : : : :*conclusions". This sentence is not necessary here, so suggest to rewrite or delete it.

**Response:** Thank you for your suggestion. We have deleted these sentences.

3. Lines 287-289: "These human activities    present study". As you discussed, human activities (reservoir construction, irrigation, sewage discharge) could disturb streamflow and conductivity. In my opinion, these activities will obviously change the original negative power function relationship between conductivity and streamflow. Therefore, it is possible to determine whether it is affected by human activities by analyzing the correlation between conductivity and streamflow. In fact, you have explained in Lines 173-174 that the negative correlation between streamflow and conductivity of the basins used in this study is less than -0.5, in other words, you have excluded those basins that are obviously affected by human activities. Therefore, I suggest to rewrite the sentence of Lines 287-289 to clearly explain the impact of human activities.

**Response:** Thank you very much for your suggestion. When we selected the research basins, we have indeed excluded basins that are obviously affected by human activities. We have rewritten the related statements, see **Page 16: Lines: 292-296**.

4. Lines 332-336: "high-salinity deep circulating groundwater", "low-salinity groundwater", "high-salinity surface water". What is the relationship between salinity and conductivity? There is no clear explanation in the article, may confuse readers. Therefore, I suggest that the "salinity" in the text should be replaced by "conductivity", including the abstract, Figure 7, and conclusion. Or explain the relationship between conductivity and salinity in detail at an appropriate place.

**Response:** Thanks for your suggestion. Generally speaking, the conductivity of streamflow is positively related to the content of TDS or chloride, which means that the more salt in the water, the greater the conductivity. Considering that the word "salinity" may cause confusion, we have replaced all "salinity" in the manuscript with "conductivity".

5. The Figures should be replaced by more clearer pictures.

**Response:** The original resolution of all pictures is greater than 600×600dpi, and we will upload these original pictures in subsequent submissions.

6. There are some problems in spelling, grammar, expression and format.

**Response:** The manuscript we submitted has been polished by a professional editor whose native language is English. If there are still problems, we can check it again later.

**Reference**

Bishop, K., Seibert, J., Köhler, S., and Laudon, H.: Resolving the Double Paradox of rapidly mobilized old water with highly variable responses in runoff chemistry, Hydrological Processes, 18, 185-189, 10.1002/hyp.5209, 2004.

Cartwright, I., and Morgenstern, U.: Using tritium and other geochemical tracers to address the "old water paradox" in headwater catchments, Journal of Hydrology, 563, 13-21, 10.1016/j.jhydrol.2018.05.060, 2018.

Kienzler, P. M., and Naef, F.: Subsurface storm flow formation at different hillslopes and implications for the 'old water paradox', Hydrological Processes, 22, 104-116, 10.1002/hyp.6687, 2008.

Kirchner, J. W.: A double paradox in catchment hydrology and geochemistry, Hydrological Processes, 17, 871-874, 10.1002/hyp.5108, 2003.

---

## Referee Comment (RC2) · Ian Cartwright (Referee) · 11 Jan 2021

Ian Cartwright (Referee)

ian.cartwright@monash.edu

Understanding baseflow is important and not straightforward and papers such as this which compare techniques are valuable. This is an interesting paper that I consider is publishable following moderate revisions. I have made several comments below that I hope are helpful. The Conclusions are a little understated and for more impact, I suggest that the authors explain better what is new and useful here. Perhaps because I thought that the Eckhardt filter always would yield different information to the chemical mass balance, I was not surprised that it is not calibratable in this was at least on a daily timestep (although many studies still seem to try).

Units. Flow is reported in cubic feet per second. I realise that the USA uses imperial units but SI units are preferable.

Although not suggesting that it needs to be included, the just published paper by Cartwright & Miller (2021, Journal of Hydrology, 593, 125895, https://doi.org/10.1016/j.jhydrol.2020.125895) also looks at the variability of stream EC and implications for water stores that sustain streamflow.

Specific Comments

Abstract

The abstract is a good summary of the paper, but as with the conclusions it could be more impactful

Introduction

Line 27: Should be rainfall not rainfall-runoff

Lines 34-39. It would be useful to expand on this. Baseflow does indeed contain regional groundwater but also contains the other delayed stores of water that you discussed earlier (interflow, bank storage and return waters, slowly draining pools on the floodplain). Many of the earlier papers in hydrograph separation (eg Nathan and McMahon, 1990) do not necessarily equate baseflow with groundwater inflows (see the first paragraph of their conclusions); however, many of the papers that have applied these techniques have. This is a subtle but important point whereby the assumptions with these techniques have changed a little over time.

Lines 60-64. Benchmarking baseflow separation methods is difficult due to the associated assumptions (especially that we are applying a two-component separation to a multi-component system). By what criteria did Xie et al. determine this? Perhaps it is best to leave out that statement and concentrate on the uncertainties. In practice because the Eckhardt method is "tuneable" it should always perform batter than the other filters with fixed parameter; however, that does not overcome the fundamental

problems with this approach.

Lines 70-74. The CMB approach goes back before Stewart et al. (2007). I think that Pinder & Jones (1969, Water Resources Research, 5, 438-445) introduced the technique and Yu & Schwartz (1999, Hydrological Processes, 13, 191-209) further formulated it.

Lines 75-95. All this is correct, but in the context of this paper you should think about what is important. You already have mentioned that rivers contain water from multiple sources, in which case two-component hydrograph separations are not ideal. So is it more likely that the comparison will reveal those intermediate water stores (as in the Cartwright et al. and Rammal et al. studies) rather that being a viable method to calibrate the BFI parameter. There is also a timescale issue here. It may be that the BFI parameter can be calibrated on a long timescale (seasonal or annual) but not on a daily or weekly timescale (i.e., is it an annual average baseflow or a daily baseflow that you are concerned with?).

Methods

Lines 131. See comment above about the origins of the CBM method.

Lines 131-135. Somewhere here or in the introduction you should mention that the conductivity is presumed to reflect the overall salinity or concentration of a conservative component (e.g., Cl).

Lines 136-141. Assigning the baseflow EC as the maximum (or 99th percentile) value assumes that at low flows the river is entirely fed by baseflow. This is probably fine as an assumption in drier areas but may not always be the case in high-rainfall areas. In many areas this maximum value is lower than the EC of regional groundwater – which is one of the lines or evidence that near-river water stores (such and bank return flows) may always contribute to the river (McCallum et al., 2010 and Cartwright & Irvine, 2020 both discuss this). This is also worth a brief discussion here.

Also did you assign a constant baseflow to each water year (or the whole record) or use the strategy outlined in Miller's papers where they interpolate between the maximum EC in each water year to assign a value of baseflow EC on individual days?

Lines 146-151. There is some repetition here with the introduction. Since this is the methods, just tell us how you did the calibration.

It would be useful in include a bit of QC information. How complete are the records and did you attempt to infill missing data?

Results

In addition to the constraints described above. Baseflow estimation based on hydrograph separation requires that the flow regime is not overly influenced by human activities (eg major dams or storages on the river). Both baseflow estimation techniques methods are best applies to streams that are uniformly gaining (both along their reaches and at all times). Can you be more specific as to whether the streams met these criteria.

Looking at Fig. 3, there is some difference with the results of Cartwright et al. (2014) in as much as there was a seasonal difference in that study – the estimates agreed better in summer than winter (proposed as being due to a higher proportion of transient water stores in winter). Do you see that in any of your studies?

Discussion

Line 240. Not sure what you mean by converge

Lines 243-246. Do you know whether that is really the case in your catchments? If there are the data it would be interesting to know whether the salinity of the stream ever reaches that of the regional groundwater as that informs us whether the transient water stores ever truly are absent.

Lines 270-275. It would be useful to briefly introduce hysteresis loops and the informa-

tion that they provide in the introduction.

Lines 283-290. Are any of these basins severely impacted by human activities and what steps did you take to exclude basins that might be unsuitable?

Lines 326-336 and Fig. 7. This works well as a general concept but I would just call the "low salinity groundwater" something like "low salinity transient water" to be consistent with the way that you have discussed it in the paper. Some of that input is from the saturated zone (eg the bank return flow) so is groundwater but there may also be interflow or water from floodplain pools here.

Conclusions

These are a little understated. It is not surprising, given the assumptions inherent in the two techniques and the previous work that there is disagreement. Many of these conclusions have been made before by the studies that you quote earlier. So what is new and important here? Is it possible to use the calibration of the Eckhardt method to estimate total annual baseflow and then use the differences to do multicomponent separation? Does your study help understand the timescales over which either or both techniques yield useful information? Are there river types (size, rainfall, topography) where the comparison worked better?

Strengthening the conclusions would give the paper more impact.

---

## Referee Comment (RC3) · Ian Cartwright (Referee) · 12 Jan 2021

This is something that occurred to me after submitting my review that I have not seen much discussion of. One difference between the CMB method and other methods based on the hydrograph (eg Eckhardt or the local minima techniques) is the number of times in a year that the stream is conceived to be fed entirely by baseflow. Assigning a single value for EC(BF) based on the highest stream EC generally means that the calculated baseflow is 100% only on one or two days at low summer flows. However, the hydrograph techniques generally predict that the stream is fed by baseflow multiple times in the year between the high flow peaks. This difference in conceptualisation

probably relates to how the techniques apportion the waters (so with the CMB, the baseflow component may be mostly saline groundwater and the hydrograph techniques may be grouping all delayed water stores as baseflow). If changing BFI in the Eckhardt method only results in a decrease in the volume of baseflow but not the frequency of when the steam is estimated to be dominated by baseflow, there will always be disagreement on short timescales. Perhaps this is worth commenting on.

---

## Author Comment (AC2) · 29 Jan 2021

Thank you very much for your additional explanation. We fully agree with the points you discussed here. The basic assumptions of the CMB method and the Eckhardt method cause them to get significantly different full baseflow segments during the dry season. The calibration process of BFImax will not improve this difference. In fact, the calibration is likely to aggravate the deviation of the two methods in the dry season (Fig. S5). We have added relevant discussions in the manuscript.

---

## Author Comment (AC3) · 29 Jan 2021

**Response to reviewer #2**

*Explanation: The reviewer' comments are shown in black, the author's replies and revises are shown in blue. The quoted line numbers and page numbers are the Marked manuscripts.*

**Comments from reviewer #2:**

Understanding baseflow is important and not straightforward and papers such as this which compare techniques are valuable. This is an interesting paper that I consider is publishable following moderate revisions. I have made several comments below that I hope are helpful. The Conclusions are a little understated and for more impact, I suggest that the authors explain better what is new and useful here. Perhaps because I thought that the Eckhardt filter always would yield different information to the chemical mass balance, I was not surprised that it is not calibratable in this was at least on a daily timestep (although many studies still seem to try).

**Response:** Thank you very much for reviewing our manuscript. We have revised the manuscript based on your comments and rewritten the conclusion part to emphasize the important findings and significance of this research. See the following for the reply to each comment.

Units. Flow is reported in cubic feet per second. I realise that the USA uses imperial units but SI units are preferable.

**Response:** Thank you for your reminder. We have converted all the units in the manuscript to the SI unit system, including Fig.3, Fig.5- Fig.7, Fig.S1- Fig.S5.

Although not suggesting that it needs to be included, the just published paper by Cartwright & Miller (2021, Journal of Hydrology, 593, 125895, https://doi.org/10.1016/j.jhydrol.2020.125895) also looks at the variability of stream EC and implications for water stores that sustain streamflow.

**Response:** Thank you for your recommendation, which gives us the privilege to read this interesting and highly relevant article. We have carefully read this article and introduced its views and main conclusions into our manuscript.

**Specific Comments:**
**Abstract:**

The abstract is a good summary of the paper, but as with the conclusions it could be more impactful.

**Response:** Thanks, we have re-adjusted the summary based on the new conclusions.

*Introduction:*

Line 27: Should be rainfall not rainfall-runoff.

**Response:** Okay, it has been modified.

Lines 34-39. It would be useful to expand on this. Baseflow does indeed contain regional groundwater but also contains the other delayed stores of water that you discussed earlier (interflow, bank storage and return waters, slowly draining pools on the floodplain). Many of the earlier papers in hydrograph separation (eg. Nathan and McMahon, 1990) do not necessarily equate baseflow with groundwater inflows (see the first paragraph of their conclusions); however, many of the papers that have applied these techniques have. This is a subtle but important point whereby the assumptions with these techniques have changed a little over time.

**Response:** Thank you for your suggestion. We have discussed the content of this part in detail and emphasized the minor change in the concept of baseflow.

Lines 60-64. Benchmarking baseflow separation methods is difficult due to the associated assumptions (especially that we are applying a two-component separation to a multi-component system). By what criteria did Xie et al. determine this? Perhaps it is best to leave out that statement and concentrate on the uncertainties. In practice because the Eckhardt method is "tuneable" it should always perform better than the other filters with fixed parameter; however, that does not overcome the fundamental problems with this approach.

**Response:** Thanks for your suggestion, we have re-read the article by Xie et al. Their criterion is to select the most likely whole baseflow segments according to the recession theory, and then use these segments to test the effect of different baseflow separation methods. In our opinion, this criterion is based on the basic assumption of linear reservoir theory, that is to say, it is still empirical. Therefore, we agree with your point of view that the Eckhardt method has obvious advantages over other filtering methods or graphical methods, but its essence is still a filtering method, we can hardly say that it has the best effect, so we deleted this sentence.

Lines 70-74. The CMB approach goes back before Stewart et al. (2007). I think that Pinder & Jones (1969, Water Resources Research, 5, 438-445) introduced the technique and Yu & Schwartz (1999, Hydrological Processes, 13, 191-209) further formulated it.

**Response:** Thank you for your reminder. We have carefully read these articles and traced the CMB method back to these studies.

Lines 75-95. All this is correct, but in the context of this paper you should think about what is important. You already have mentioned that rivers contain water from multiple sources, in which case two-component hydrograph separations are not ideal. So is it more likely that the comparison will reveal those intermediate water stores (as in the Cartwright et al. and Rammal et al. studies) rather that being a viable method to calibrate the BFI parameter. There is also a timescale issue here. It may be that the BFI parameter can be calibrated on a long timescale (seasonal or annual) but not on a daily or weekly timescale (i.e., is it an annual average baseflow or a daily baseflow that you are concerned with?).

**Response:** Thank you for your suggestion, we agree with your opinion. We have adjusted the discussion in this paragraph according to the background of the article and emphasized the importance of comparing different methods to determine multiple transient water sources.

**Methods:**

Lines 131. See comment above about the origins of the CBM method.

**Response:** We have revised the introduction about the origin of the CMB method.

Lines 131-135. Somewhere here or in the introduction you should mention that the conductivity is presumed to reflect the overall salinity or concentration of a conservative component (e.g., Cl).

**Response:** Okay, we have discussed the relationship between conductivity and salinity or conservative components in the methods section.

Lines 136-141. Assigning the baseflow EC as the maximum (or 99th percentile) value assumes that at low flows the river is entirely fed by baseflow. This is probably fine as an assumption in drier areas but may not always be the case in high-rainfall areas. In many areas this maximum value is lower than the EC of regional groundwater – which is one of the lines or evidence that near-river water stores (such and bank return flows) may always contribute to the river (McCallum et al., 2010 and Cartwright & Irvine, 2020 both discuss this). This is also worth a brief discussion here.

**Response:** Okay, we have discussed the possible underestimation and the reasons for using the maximum streamflow conductivity to estimate the conductivity of the baseflow in high-rainfall areas.

Also did you assign a constant baseflow to each water year (or the whole record) or use the strategy outlined in Miller's papers where they interpolate between the maximum EC in each water year to assign a value of baseflow EC on individual days?

**Response:** Yes, we use Miller's strategy, which has been pointed out in the article.

Lines 146-151. There is some repetition here with the introduction. Since this is the methods, just tell us how you did the calibration.

**Response:** OK, we have abbreviated these sentences to avoid too much repetition.

It would be useful in include a bit of $Q_C$ information. How complete are the records and did you attempt to infill missing data?

**Response:** Thanks for your suggestion, we have added the completeness of streamflow conductivity and interpolation principle in the data section. The streamflow data of all stations are continuous and complete, and the complete rate of the conductivity data of each station is greater than 90%. For the missing data, this study performed linear interpolation based on the conductivity values at both ends of the missing period.

**Results:**

In addition to the constraints described above. Baseflow estimation based on hydrograph separation requires that the flow regime is not overly influenced by human activities (eg. major dams or storages on the river). Both baseflow estimation techniques methods are best applies to streams that are uniformly gaining (both along their reaches and at all times). Can you be more specific as to whether the streams met these criteria.

**Response:** Thanks for your suggestion, we have discussed the basis for selecting these basins in the data section. We use the negative correlation coefficient between conductivity and runoff to exclude those hydrological stations that are obviously affected by human activities such as reservoirs, similar to the study of Miller et al., (2014).

Looking at Fig. 3, there is some difference with the results of Cartwright et al. (2014) in as much as there was a seasonal difference in that study – the estimates agreed better in summer than winter (proposed as being due to a higher proportion of transient water stores in winter). Do you see that in any of your studies?

**Response:** What needs to be explained here is that the periodic deviation mentioned in this study is after calibration, and the focus is on the difference between the rising and falling limbs of the wet season. It does not focus on the difference between wet and dry seasons. In terms of wet season and dry season, the deviation range of dry season between two wet seasons is often smaller (Fig. 3, Fig. S1-S5), as discussed by Cartwright et al. (2014), which may be mainly due to more transient water source components in the streamflow during wet season. In addition, the calibration process may aggravate the deviation of the dry season, Fig. S5 is an obvious example. Before calibration, the separation results of the two methods in the dry season are basically the same. After calibration, although the total deviation is minimized, the deviation in the dry season is significantly increased, which is contrary to common sense. We have added relevant discussions in the results section of the manuscript.

**Discussion:**

Line 240. Not sure what you mean by converge.

**Response:** We have changed the expression of this sentence. Our original intention is that transient water sources will flow into the river at different times and places.

Lines 243-246. Do you know whether that is really the case in your catchments? If there are the data it would be interesting to know whether the salinity of the stream ever reaches that of the regional groundwater as that informs us whether the transient water stores ever truly are absent.

**Response:** Indeed, we cannot be sure that every river basin will achieve the described situation, this is just a phenomenon that may exist in most river basins. Due to the lack of groundwater conductivity data, we have not been able to determine when the transient water sources will end. So we adjusted the expression of these sentences to make them more rigorous.

Lines 270-275. It would be useful to briefly introduce hysteresis loops and the information that they provide in the introduction.

**Response:** Okay, we have discussed the hysteresis loops and the information it reflects in the introduction.

Lines 283-290. Are any of these basins severely impacted by human activities and what steps did you take to exclude basins that might be unsuitable?

**Response:** We have discussed the basis for selecting these basins in the data section. We use the negative correlation coefficient between conductivity and streamflow to exclude those hydrological stations that are obviously affected by human activities such as reservoirs.

Lines 326-336 and Fig. 7. This works well as a general concept but I would just call the "low salinity groundwater" something like "low salinity transient water" to be consistent with the way that you have discussed it in the paper. Some of that input is from the saturated zone (eg the bank return flow) so is groundwater but there may also be interflow or water from floodplain pools here.

**Response:** Thank you for your suggestions. We have changed these nomenclatures based on your suggestions to ensure their accuracy.

**Conclusions:**

These are a little understated. It is not surprising, given the assumptions inherent in the two techniques and the previous work that there is disagreement. Many of these conclusions have been made before by the studies that you quote earlier. So what is new and important here? Is it possible to use the calibration of the Eckhardt method to estimate total annual baseflow and then use the differences to do multicomponent separation? Does your study help understand the timescales over which either or both techniques yield useful information? Are there river types (size, rainfall, topography) where the comparison worked better?

Strengthening the conclusions would give the paper more impact.

**Response:** Thank you for your suggestions. We have adjusted the structure and presentation of the conclusion section, emphasized the innovations and main contributions of this research, the precautions for the application of the two baseflow separation methods, and the enlightenment for future research. The following is a reply to the four questions you raised:

*For the first question:* This study attempts to solve the confusion caused by the application of these two baseflow methods from a new perspective, focusing on the calibration effect of the CMB method on the Eckhardt method on the daily scale. In addition, this study validated and expanded the previous research conclusions (McCallum et al., 2010; Cartwright et al., 2014) within a wider range of watershed characteristics, and clarified the concept of four-component streamflow separation by comparing the results of the two methods.

*For the second question:* It is unwise to try to use the CMB method to calibrate the Eckhardt method. Instead, future research should optimize the determination criteria of the parameters based on the basic assumptions of the respective methods, and then achieve multi-component separation through comparison.

*For the third question:* If the baseflow is defined as the discharge amount of regional groundwater to river (which is used to evaluate the repeated calculation amount between surface water and groundwater resources), then the CMB method will have large errors in the rising limb, while the Eckhardt method will have large errors in the recession limb. Therefore, it may be a more reasonable choice to use Eckhardt method in the rising limb and CMB method in the falling limb.

*For the fourth question:* The main content of this study is to analyze the calibration effect of the CMB method on the Eckhardt method, so it only discusses the possibility of achieving the four-component separation of streamflow through comparison. In addition, we do not have detailed information on the characteristics of the 26 basins, so we cannot quantitatively analyze which basin type is more suitable for comparison. We will focus on the comparative effects under different basin characteristics in future research.